



Atmospheric
Measurement
Techniques

# Ice crystal number concentration from lidar, cloud radar and radar wind profiler measurements

**Johannes Bühl, Patric Seifert, Martin Radenz, Holger Baars, and Albert Ansmann**

Leibniz Institute for Tropospheric Research, Permoserstr. 15, 04318 Leipzig, Germany

**Correspondence:** Johannes Bühl (buehl@tropos.de)

**Abstract.** A new method for the retrieval of ice crystal number concentration (ICNC) from combined active remote-sensing measurements of Raman lidar, cloud radar and radar wind profiler is presented. We exploit – for the first time – measurements of terminal fall velocity together with the radar reflectivity factor and/or the lidar-derived particle extinction coefficient in clouds for retrieving the number concentration of pristine ice particles with presumed particle shapes. A lookup table approach for the retrieval of the properties of the particle size distribution from observed parameters is presented. Analysis of methodological uncertainties and error propagation is performed, which shows that a retrieval of ice particle number concentration based on terminal fall velocity is possible within 1 order of magnitude. Comparison between a retrieval of the number concentration based on terminal fall velocity on the one hand and lidar and cloud radar on the other shows agreement within the uncertainties of the retrieval.

## 1 Introduction

Aerosols, clouds and precipitation are major components of Earth's climate system. The complex aerosol–cloud-dynamics interaction currently poses major challenges for the numerical modeling of climate and weather phenomena because the majority of rain formation on Earth happens through the ice phase (Mülmenstädt et al., 2015). The process of heterogeneous ice nucleation in clouds is of particular importance because it constitutes the link between aerosol conditions – including ice-nucleating particle concentration (IPNC) – and precipitation formation. An understanding of ice nucleation and growth is necessary for understanding precipitation formation, cloud stability (Korolev et al., 2017), secondary ice formation (Sullivan et al., 2017) and cloud radiative transfer (Sun and Shine, 1994). It is, hence, a key process for the global weather and climate system which must be understood in detail in order to make accurate predictions about cloud and precipitation patterns in state-of-the-art numerical weather forecasts and future climate projections.

To date, ice nucleation has not been able to be observed directly in the atmosphere, but we are gaining the ability to retrieve ice crystal number concentration (ICNC, further designated as $N$) and the respective ice crystal number flux ($F$). Both are promising approaches to gain quantitative information about ice nucleation in clouds. Apart from $N$, $F = N \times v_t$ (with $v_t$ as the terminal fall velocity) is especially well suited to derive the rate of ice production in clouds. An illustration of the use of $F$ in comparison with $N$ is given in Fig. 1. $F$ of falling ice particles yields a direct measure of the rate of ice production in the cloud above. Bühl et al. (2016) estimated ice mass fluxes produced in well-constrained shallow stratiform cloud layers. Based on these measurements, information about the contribution of ice precipitation on the mass balance of the mixed-phase cloud layer was retrieved.

Aircraft observations have been used frequently for measuring $N$ and $F$ via optical measurement of the particle size distribution (PSD) of ice crystals (Eidhammer et al., 2010; Westbrook and Illingworth, 2013; Voigt et al., 2017). Such observations can indeed deliver a quantitative picture of $N$ and the shape of particles, but since they take place at only one height level, the actual level of ice formation is often not known, thus blurring the resulting long-term statistics. Ground-based remote sensing provides accurate information

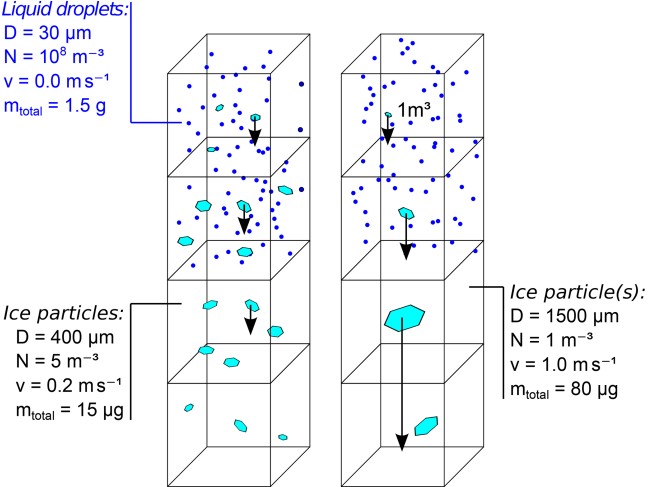

Liquid droplets:
$D = 30\,\mu m$
$N = 10^8\,m^{-3}$
$v = 0.0\,m\,s^{-1}$
$m_{total} = 1.5\,g$

$1\,m^3$

Ice particles:
$D = 400\,\mu m$
$N = 5\,m^{-3}$
$v = 0.2\,m\,s^{-1}$
$m_{total} = 15\,\mu g$

Ice particle(s):
$D = 1500\,\mu m$
$N = 1\,m^{-3}$
$v = 1.0\,m\,s^{-1}$
$m_{total} = 80\,\mu g$

**Figure 1.** Comparison of two clouds with different ice crystal number concentrations $N$ but with the same ice crystal number flux, $F = 1\,m^{-2}\,s^{-1}$ ($F = N \times v_t$ with $v_t$ terminal fall velocity), and, hence, also the same rate of ice production: **(a)** higher ice crystal number concentration ($N$) but less total ice mass and **(b)** lower $N$ and higher total ice mass.

about the ice nucleation level. However, retrieving $N$ from remote-sensing measurements is extremely challenging. In particular, freshly created pristine ice crystals pose challenges in this context because they vary strongly in shape over different ranges of ice nucleation temperatures, thus sensitively influencing the accuracy of retrievals and model results (Simmel et al., 2015).

Today, all remote-sensing approaches for retrieving $N$ need a priori information about crystal size before it is possible to be able to derive $N$. Extensive observational variables like lidar-derived optical particle extinction $E$ or radar reflectivity factor $Z$ can then be used to retrieve $N$. Hence, the task of deriving an estimation of particle size is central to deriving $N$. Different methods exist for retrieving a proxy for particle size. Mitchell et al. (2018) use a combination of active and passive remote-sensing sensors in order to constrain the properties of the observed cloud particles. Cazenave et al. (2019), Delanoë and Hogan (2010), and Sourdeval et al. (2018) employ a forward-iteration method in order to obtain an estimation of $N$ from combined observations of spaceborne lidar and cloud radar. Employing these techniques, an estimation of $N$ based on assumptions (e.g., on particle shape) is well constrained if both lidar and cloud radar (CR; Görsdorf et al., 2015) observations are available. This method exploits the strong wavelength dependence in the efficiency of the backscatter signal between lidar (geometrical scattering) and radar (Rayleigh/Mie scattering) for ice particles. In optically thick clouds, where only CR measurements are available, the method falls back to parameterizations of particle size, and the retrieval of $N$ is no longer possible. Also multi-frequency radar observations have been

introduced for the measurement of particle sizes (Battaglia et al., 2014; Sekelsky et al., 1999; Kneifel et al., 2011). However, such methods do not work for pristine particles (with a diameter smaller than about 1 mm, Battaglia et al., 2014). For these particles, only the terminal fall velocity $v_t$ leaves traces about particle size, but actual observations of $v_t$ are difficult to obtain. Such observations have been made in laboratories (Fukuta and Takahashi, 1999) and, recently, Bühl et al. (2015) and Radenz et al. (2018) showed that a combination of the CR and radar wind profiler (RWP; Steinhagen et al., 1998; Lehmann and Teschke, 2001) can be used in order to derive $v_t$ of ice particles.

In the present work, we make use of these new measurements and present an alternative approach for the retrieval of $N$ that works in the presence of very small pristine ice crystals and in clouds which cannot be penetrated by lidar. $v_t$ and spectral width $w$ obtained from combined Doppler spectra observed with the CR and RWP are used to derive the median diameter ($D_m$) and the shape parameter $\mu$ of a gamma-$\mu$ distribution. This kind of distribution is mentioned and shortly discussed in Delanoë et al. (2005) as one option for a universally applicable PSD for ice crystal populations. A forward model approach based on a lookup table of numerically derived microphysical and observed parameters is employed in the present work. Combinations of environmental and microphysical input parameters are used to compute a lookup table with a set of observable variables, e.g., $v_t$, $E$ and $Z$. The observable variables are compared to the actually measured variables in order to come up with an estimation of ice crystal properties, $F$ and $N$. We concentrate on the description and evaluation of the method and present one case study in which different ways of performing the retrieval are exercised.

The paper is structured as follows. Section 2 describes how the data used in the paper has been acquired. In Sect. 3 the method for deriving information about the PSD from both $v_t$ and the combined lidar/radar measurements is described. Section 4 presents example case studies. The conclusion and outlook are given in Sect. 5.

## 2 Data

The remote-sensing data used in the context of this work were acquired during the COLRAWI-2 campaign (combined observations with lidar radar and wind profiler) at the Richard Assmann Observatory (RAO) of the Deutscher Wetterdienst in Lindenberg, Germany, between 1 June and 30 September 2015 (Bühl et al., 2015). Figure 2 shows a case study of the combined measurements at Lindenberg. During the project, an ultra-high-frequency (UHF) RWP was used to measure the vertical velocity of air ($v_{air}$) in combination with measurements of $Z$, the mean Doppler velocity $v_D$ and spectral width $w$ from a vertically pointing 35 GHz CR. The RWP was switching between vertically pointing and horizontal beam swinging every 30 min. This special observa-

tion mode of the RWP was necessary to acquire direct measurements of both horizontal and vertical air motions with only one RWP available. A Polly$^{XT}$ Raman lidar (Engelmann et al., 2016) pointing 5° off zenith was used for the measurement of the ice particle extinction coefficient ($E$) which is derived at a wavelength of 1064 nm (Baars et al., 2017). For the sake of completeness also a Stream Line XR Doppler lidar (Päschke et al., 2015) should be mentioned which was also observing vertical Doppler velocity $v_{DL}$. However, its measurements are left out of the retrieval method because they (a) provide mainly redundant information to the cloud radar and (b) pose additional problems in the interpretation of optical signals due to specular reflection at the planar planes of horizontally oriented ice crystals.

A Cloudnet (Illingworth et al., 2007) dataset was derived from the lidar/CR/MWR measurements including the attenuation-corrected values of CR reflectivity used in this paper. The combined measurements of CR and RWP were used to derive a dataset of a particle ensemble mean $v_t$ with an error margin of about 0.1 m s$^{-1}$ (Radenz et al., 2018). It is worth noting here that in the context of this work we use "uncertainty" to describe retrieval and methodological uncertainties and "errors" for measurement errors. This unique dataset is used here for the first time to test the retrieval method and give examples.

## 3 Methods

### 3.1 Measuring terminal fall velocity

The retrieval presented in this paper is based on measurements of radar reflectivity factor $Z$ and terminal fall velocity $v_t$ with CR and RWP. Measuring $Z$ and $v_t$ of ice particles in clouds is difficult because different factors influence the CR Doppler spectrum of ice particles.

- Vertical air motions shift the Doppler spectrum in such a way that $v_D = v_{air} + v_t$.

- Turbulence and beam width effects broaden the Doppler spectrum (Shupe et al., 2008).

The shift induced by vertical air motions can be removed if the magnitude of the vertical air motion $v_{air}$ is known. In Radenz et al. (2018) a method is presented for measuring $v_t$ and $v_{air}$ with a combination of CR and RWP. The method of Radenz et al. (2018) combines Doppler spectra observed by both instruments in order to remove the influence of Rayleigh scattering on the otherwise Bragg-scattering-dominated RWP measurements, resulting in an undisturbed measurement of vertical air motions. The velocity scale of the CR Doppler spectrum is shifted by the measured vertical air motion in order to derive $v_t$. In the context of this work, $v_t$ is of central importance for the retrieval of particle size. The proxy for particle size is the particle maximum diameter $D$

(see Mitchell, 1996) which describes the diameter of a sphere just encircling the total ice crystal.

Mie scattering effects are neglected in the context of this work because we aim on studying pristine ice particles. Signal attenuation by these ice particles is also negligible. Turbulence and beam width broadening are also introducing artifacts. A strongly broadened CR Doppler spectrum might contain unphysical negative terminal fall velocities even after the correction of mean vertical air motion. Such effects cannot be removed easily, but luckily they only affect the width and not the mean velocity of the spectrum, which is shown in Sect. 3.2.

### 3.2 General principle of the retrieval method

In the present work, a method for deriving $N$ and $F$ of pristine ice particles from combined lidar, CR and RWP measurements is described. An analytical inversion of the measurement values of lidar, CR and RWP is not possible, so numerical inversion techniques have to be applied. For efficient and simple numerical implementation, a lookup table is used which contains the properties of the PSD and the observable measurement values $Z$, $E$, $v_t$ and CR spectral width $w$. In this section, the mathematical foundations and assumptions for creating this lookup table are explained.

The basic measurement values that are used in the retrieval are the first three moments ($Z$, $v_t$ and $w$), and they are derived from the CR Doppler spectra which are corrected for vertical air motion. If no RWP measurements are available, $v_t$ can be replaced by the ratio $Z/E$, with $E$ being the particle extinction coefficient measured by lidar. A retrieval is performed by trying to find a particle distribution that leads to the same variables as the measured ones. Figure 3 gives an example of this forward modeling approach. Two PSDs, characterized by different particle shapes of "side planes" and "column-like particles", respectively, are defined in such a way that the simulated CR Doppler spectrum matches with the one measured. It is visible from this figure that different particle shapes can lead to very similar cloud radar spectra from significantly different PSDs. This is the case because the relationship between mass and terminal fall velocity are different for both particle populations side planes and column-like. For retrieving $N$ and $F$, the simulation procedure is done first with a large variety of size distributions which are later compared to the measured values. A schematic overview of the retrieval method used here is given in Fig. 4.

Environmental factors affecting the shape of the CR Doppler spectra are taken into account during the computation of the lookup table. The signal strength of the CR is, e.g., affected by attenuation from water vapor and liquid water particles; air motion shifts the CR Doppler spectrum and turbulence broadens it. In the context of this work, water vapor attenuation is corrected with the method of Cloudnet (Illingworth et al., 2007). Particle attenuation is neglected since we concentrate on cloud layers, in which ice, liquid water and

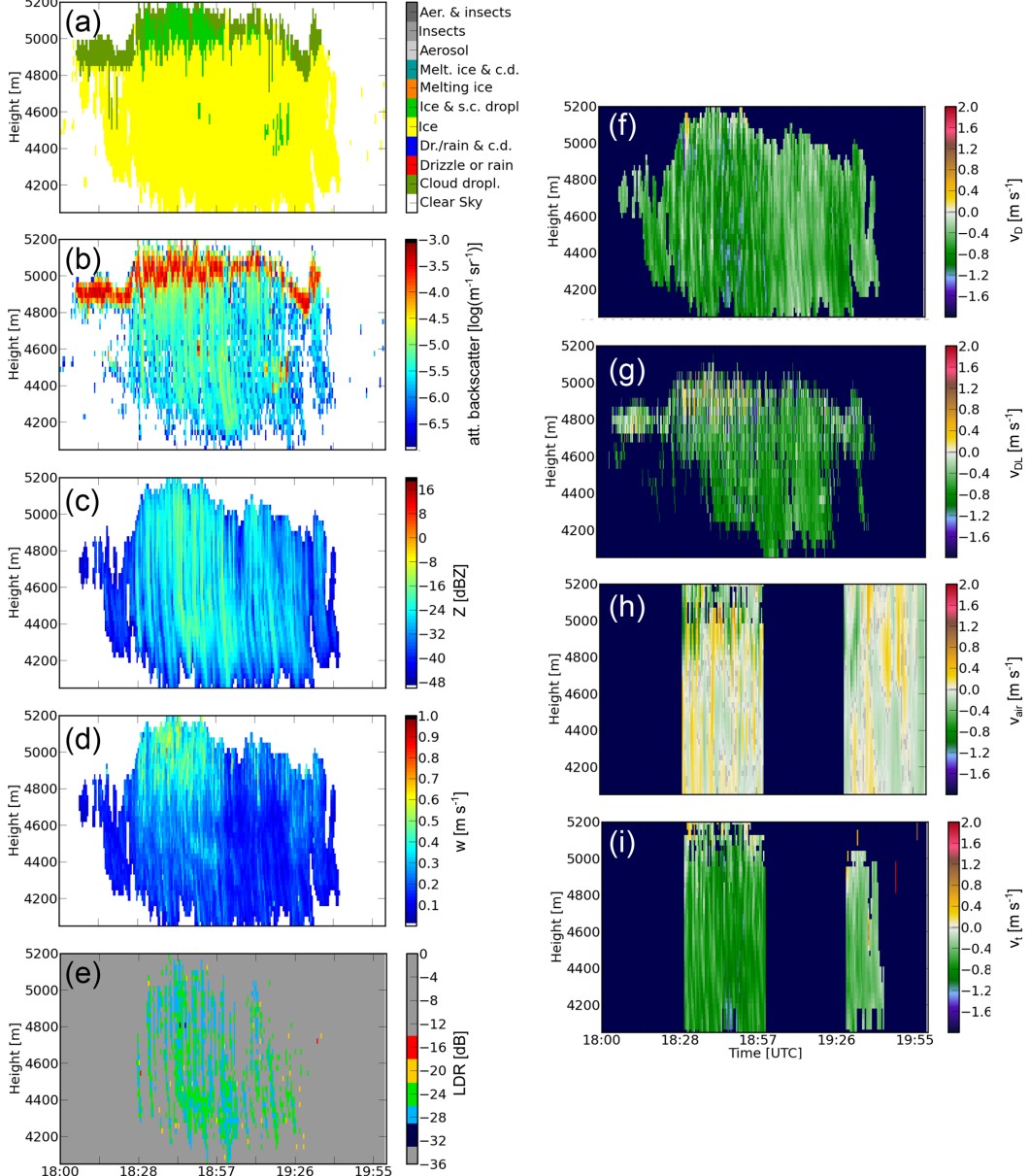

**Figure 2. (a)** Cloudnet target classification, **(b)** attenuated backscatter from lidar at the 1064 nm wavelength, **(c)** CR reflectivity $Z$, **(d)** CR Doppler spectral width $w$, **(e)** CR linear depolarization ratio (LDR), **(f)** CR Doppler velocity $v_D$, **(g)** Doppler lidar Doppler velocity $v_{DL}$, **(h)** vertical air motions as measured with the RWP $v_{air}$ and **(i)** reflectivity-weighted terminal fall velocity $v_t$ are shown for a mixed-phase cloud layer observed on 11 June 2015 at Lindenberg. $v_t$ and $v_{air}$ are only available in 30 min intervals because vertical and horizontal wind measurement modes of the RWP alternate every 30 min.

water vapor do not contribute significantly to cloud radar attenuation.

The modified gamma distribution from Delanoë et al. (2005) is assumed here, so we set for the PSD

$$N(D) = N_{total} C \left( \frac{D}{D_m} \right)^{\mu} \exp \left( -(4+\mu) \frac{D}{D_m} \right), \quad (1)$$

with size parameter $D_m$ (median particle maximum diameter), shape parameter $\mu$ (describing the tilt of the distribution)

and normalization factor $C$ and the total ice number concentration $N_{total}$. $C$ is chosen in a way so that $\int_0^\infty N(D)dD = 1\,\text{m}^{-3}$. In the context of this work, this normalization is done numerically. In consequence, all following extensive quantities are derived for a particle number concentration of 1 particle m$^{-3}$, indicated by the subscript "1".

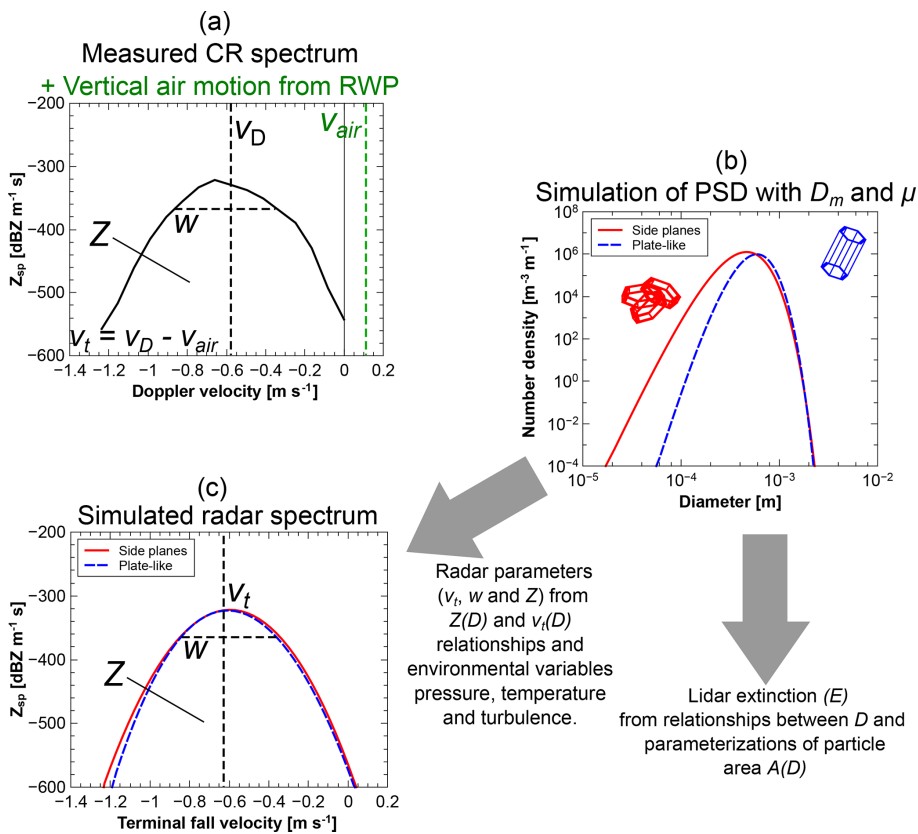

**Figure 3.** Illustration of the general idea of the retrieval. **(a)** Example CR Doppler spectrum measured at 4500 m height with the CR at Lindenberg on 11 June 2015 at 19:00 UTC (see Fig. 2) with spectral radar reflectivity ($Z_{\mathrm{sp}}(v) = Z(v)/\Delta v$ with Doppler spectrum grid length; $\Delta v = 0.08\,\mathrm{m\,s^{-1}}$) for the first three moments, $Z = \int Z(v)\mathrm{d}v$, $v_{\mathrm{t}}$ and $w$. **(b)** PSDs for two different particle populations. **(c)** Simulated $Z_{\mathrm{sp}}$ from both PSDs with the same moments as the measured variables.

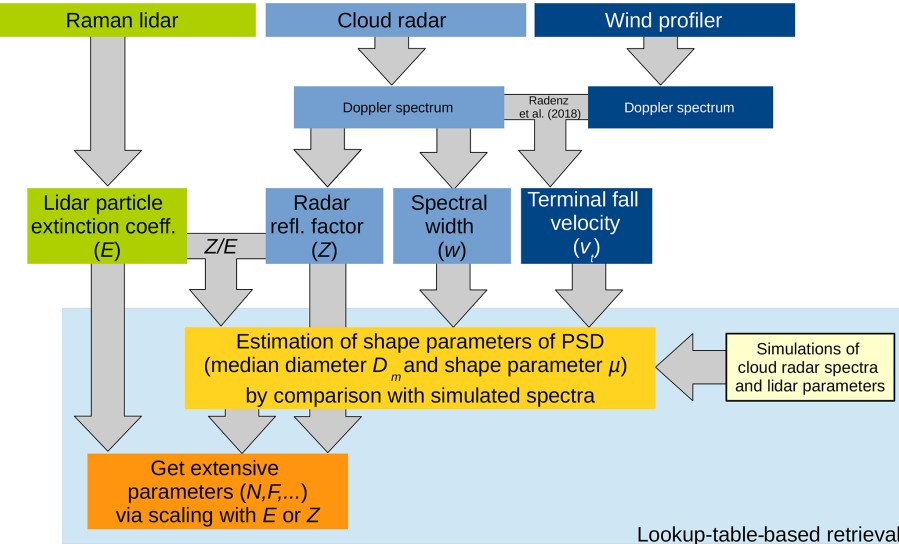

**Figure 4.** Flowchart of the retrieval algorithm, illustrating the synthesis of data from the remote-sensing instruments with the simulated parameters.

**Table 1.** Ranges and step sizes for the computation of the lookup table.

| Parameter | Range low | Range high | Step size | Unit |
|---|---|---|---|---|
| $p$ | 50 | 1050 | 50 | hPa |
| $T$ | 180 | 270 | 10 | K |
| $\sigma_{\text{total}}$ | 0.05 | 0.5 | 0.1 | $\text{m s}^{-1}$ |
| $D_{\text{m}}$ | $1 \times 10^{-5}$ | $5 \times 10^{-3}$ | $25 \times 10^{-6}$ | m |
| $\mu$ | 1 | 61 | 1 | – |

PSDs are simulated for all particle types mentioned in Table A2, and a variety of realistic combinations of the parameters pressure $p$, temperature $T$, CR spectral broadening $\sigma_{\text{total}}$, size parameter $D_{\text{m}}$ and $\mu$ are also represented. The ranges and step sizes for all parameters used in this study are found in Table 1.

The extensive properties $N$ and $F$ and the measurement quantities $Z$ and $E$ are not used as an input parameter in the forward modeling because they all linearly depend on $N$. For the retrieval, only the shape ($D_{\text{m}}$ and $\mu$) of the distribution is of interest.

A CR Doppler spectrum of $Z$,

$$Z(D) = \left( K/0.93(6/(\pi 917.0))^2 N(D)m^2(D) \right)/0.001^6,$$
(2)

is simulated with $K = 0.174$ (dielectric constant for ice at 35 GHz) for ice particles, and particle mass $m(D)$ is obtained via Eq. (B1) (Hogan et al., 2006). Only pristine particles are considered in the context of this work, and all calculations are done for a CR with 35 GHz operation frequency (8.5 mm wavelength). Since the Mie parameter of these particles is usually smaller than 1, only Rayleigh scattering is considered in this context.

Accordingly, the extensive variables normalized to 1 particle $\text{m}^{-3}$ are a normalized number flux,

$$F_1 = \int N(D)v(D)\mathrm{d}D,$$
(3)

a normalized radar reflectivity factor,

$$Z_1 = \int Z(D)\mathrm{d}D,$$
(4)

and a normalized particle extinction coefficient,

$$E_1 = 2 \times \int N(D)A(D)\mathrm{d}D,$$
(5)

with particle area $A(D)$ obtained from Eq. (B2).

From the two latter equations, the reflectivity-to-extinction ratio $Z_1/E_1 = Z/E$ is defined.

For means of completeness, the mean terminal fall velocity measured with a Doppler lidar is given as

$$v_{\text{t, DL}} = 2 \times \int N(D)A(D)v(D)\mathrm{d}D/E_1.$$
(6)

The latter formula assumes that the backscatter from all crystals is only proportional to their projected area. This requires all crystals to be either randomly oriented or aligned perpendicular to flow direction. Since this would complicate the discussion of the proof-of-concept approach presented here, we stick to cloud radar measurements of $v_{\text{t}}$. However, redundancy in observations of retrieved parameters increases the robustness of the methodology and quality of the analysis' products.

Simulation of microphysical parameters is only done over the ranges in which the particle properties are valid, which are given in Table A1. The normalized number concentration

$$N_1 = \int N(D)\mathrm{d}D$$
(7)

is computed in order to ensure that the PSD is well represented within the limits of $D$. The deviation of $N_1$ from 1 indicates how many particles are not considered due to the limited range of $D$. In the context of this work, we discard all calculations in which $N_1 < 0.95$, meaning that at maximum 5 % of the particles at the upper and/or lower boundary may be not included.

As mentioned before, a proxy for particle size is the most crucial intensive parameter for the retrieval of $N$. In the present work we exploit measurements of $v_{\text{t}}$ in order to come up with an estimation of the median diameter $D_{\text{m}}$ of the gamma-$\mu$ distribution. Hence, in this section and in Appendix B, the relationship between $D_{\text{m}}$ and $v_{\text{t}}$ is discussed. Mitchell (1996), Heymsfield and Westbrook (2010), and Khvorostyanov and Curry (2014) present an analytic theory for the calculation of $v_{\text{t}}$ of particles on the basis of four fractal parameters, describing mass and area of the particles dependent on their maximum diameter $D$ (minimum enclosing circle for ice crystals, particle diameter for droplets). The resulting formula is

$$v(D) = A_v \times D^{B_v},$$
(8)

with $A_v$ and $B_v$ also being functions of $D$ dependent on the properties of the air surrounding the falling ice crystal. The detailed derivation of the formula is described in Appendix B. Flow tunnel experiments, e.g., by Fukuta and Takahashi (1999), have produced the parameterizations of particle shapes needed for the derivation of $A_v$ and $B_v$. Tables A1 and A2 summarize these parameterizations for deriving mass $m(D)$ and area ($A(D)$) from the maximum particle diameter ($D$). The detailed derivation of the latter as well as of $v(D)$, which is used in the following, is given in Appendix B.

A CR Doppler spectrum is computed on the terminal-velocity grid by computing $v(D)$ and inverting numerically to $D(v)$. The resulting spectrum $Z(v(D))$ is artificially

broadened by numerical folding with the normalized Gaussian distribution term

$$\left(2\pi \sigma_{\text{total}}^2\right)^{-1} \times \exp\left(-0.5\frac{v^2}{\sigma_{\text{total}}^2}\right), \tag{9}$$

with the standard deviation $\sigma_{\text{total}}$, which introduces the combined effects of turbulence and beam width broadening. From this broadened spectrum $v_{\text{t}}$ (first moment) and spectral width $w$ (second central moment) are derived.

Such spectra are computed for a variety of input parameters ($p$, $T$, $\sigma_{\text{total}}$, $D_{\text{m}}$ and $\mu$), and for each combination of input variables, the input and output parameters are collected in three vectors:

- $i_{\text{L}} = (p, T, \sigma, D_{\text{m}}, \mu)_{\text{L}}$ holding the input properties that were used in the calculations,

- $p_{\text{L}} = (v_{\text{t}}, w, Z/E)_{\text{L}}$ holding the intensive properties of the distribution and

- $n_{\text{L}} = (N_1, F_1, A_1)_{\text{L}}$ containing the normalized extensive properties of the distribution,

with the subscript L indicating that these vectors are members of the lookup table. This lookup table, containing the three vectors $i$, $p$ and $n$, is computed for a set of input variables of $p$, $T$, $\sigma_{\text{total}}$, $D$ and $\mu$.

### 3.3 Using terminal fall velocity for deriving maximum particle diameter

The basis for the retrieval of $N$ is an estimation of $D_{\text{m}}$. In this section, two relationships – $v_{\text{t}}(D)$ and $Z/E(D)$ – are compared for exemplary particle types and varying properties of the underlying PSD. $v_{\text{t}}$ and $Z/E$ can both be used to derive $D_{\text{m}}$ because both are steady and differentiable with $D_{\text{m}}$. It can be seen from Fig. 5 that $v_{\text{t}}$ and $Z/E$ show a very similar dependence on $D_{\text{m}}$, which is not surprising since both are proportional to $m$ and reciprocal to $A$. For the purpose of deriving the size and shape properties of a PSD, both quantities are interchangeable, and a direct measurement of $v_{\text{t}}$ can replace a missing $Z/E$. More importantly, both $v_{\text{t}}$ and $Z/E$ only depend weakly on shape parameter $\mu$ of the PSD and the resulting width $w$ of the CR Doppler spectrum.

### 3.4 Retrieving a result from the lookup table

For retrieving $N$, an estimation of the PSD must be acquired first. This is done by comparison of measured parameters with those stored in the lookup table. An overview about the retrieval process is given in Fig. 6. Depending on which parameter is available, this procedure can be done with all combinations of available intensive parameters (currently $v_{\text{t}}$, $w$ and $Z/E$). Examples are given in Sect. 4.

*a. Retrieval and uncertainties estimation.* A vector with the measured intensive properties $m$ is matched with the entries in the lookup table in order to derive $D_{\text{m}}$ and $\mu$ of the particle distribution. The vector $m$ can be composed of any combination of $p$, $T$, $\sigma_{\text{total}}$, $v_{\text{t}}$, $w$ and $Z/E$, but in the context of this work, only $v_{\text{t}}$, $w$ and $Z/E$ are used as variational parameters for the retrieval. The matching probability of $m$ to its corresponding values in the lookup table is described by

$$P\left(p_{\text{L}}, m, e\right) = \exp\left[0.5\sum_i \left(p_{\text{L},\,i} - m_i\right)^2 / e_i^2\right], \tag{10}$$

with $p_{\text{L}}$ being the simulated properties from the lookup table, $i$ being the index running over the length of $p_{\text{L}}$, $m$ containing the corresponding actual measurement values and vector $e$ being the uncertainties assuming a Gaussian error distribution. $P$ is applied to all lines of the lookup table, and the matching probability for each single entry is found. As a result a distribution of matching probabilities is derived. At the position of the maximum of all values of $P$, the parameters of the size distribution are retrieved because they represent the best match for the input parameters. The full width half maximum of $P$ for each derived value is used to represent the uncertainty of the retrieval. Upper and lower uncertainty may differ if the distribution of $P$ is not symmetrical around its maximum. A larger uncertainty in one of the measured variables may lead to a broader distribution $P$ for the retrieved variables (see Fig. 6).

*b. Scaling.* With the shape of the size distribution known, the normalized extensive parameters $N_1$, $F_1$, $Z_1$ and $E_1$ are also known. A vector $r$ holding the results of extensive properties ($N$, $F$, $Z$ and/or $E$) is derived by multiplying all of them with the same scaling factor $S$, which can be either $S_Z = Z/Z_1$ or $S_E = E/E1$. Scaling adds the measurement uncertainty of $Z$ (CR) or $E$ (lidar).

*c. Example.* As an example, a diagram relating $F_1$ with the intensive properties $v_{\text{t}}$ and $w$ is shown in Fig. 7. The distribution of results in ($v_{\text{t}}$, $w$) space is shown. From this distribution the most probable solution (here $F_1$) is selected for measurements of $v_{\text{t}}$ and $w$. Multiplication of the selected extensive parameter with the corresponding factor $Z/Z_1$ yields the scaling factor to derive the actual $F$ and $N$. The method is here applied in a two-dimensional example, but both extensive and intensive parameters can be of an arbitrary number.

### 3.5 Estimation of cross-sensitivity of uncertainties

For estimating the uncertainties introduced by a measurement value on the retrieved quantities, the retrieval is performed for a fixed set of input parameters, and afterwards each single parameter is varied by 1 standard deviation. The errors are an estimation of the maximum measurement accuracy that can be achieved currently. Table 2 gives a comparison of the errors derived for $v_{\text{t}}$ and $Z/E$.

The measurement errors of the parameters $p$, $T$ and $\sigma_{\text{total}}$ were chosen quite large, and, anyway, they do not introduce significant errors. It can be seen from this table that for a relatively low $v_{\text{t}}$ of $0.3\,\text{m s}^{-1}$, an error of $0.1\,\text{m s}^{-1}$ in $v_{\text{t}}$ results in

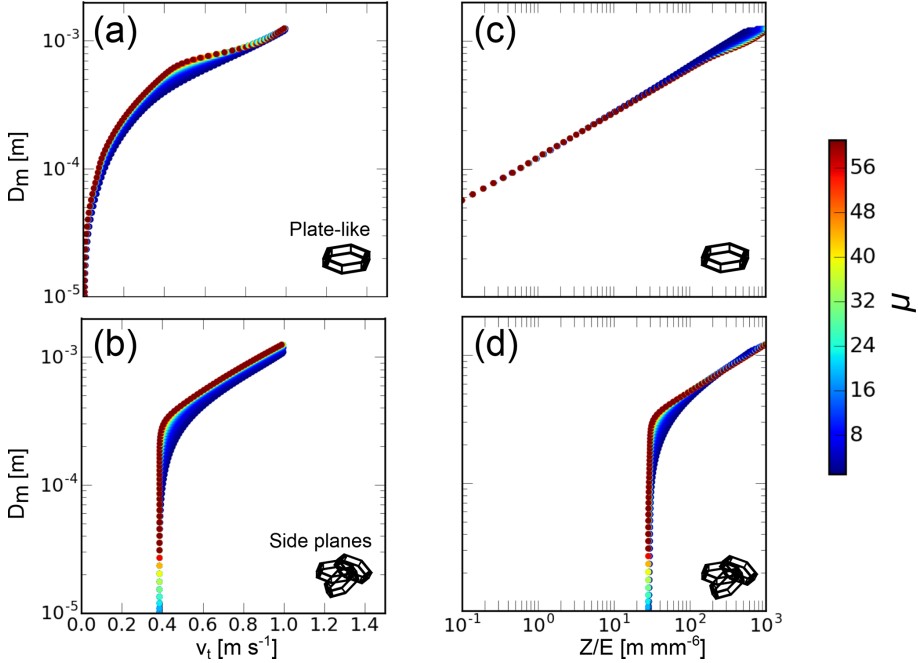

**Figure 5.** Relationship of $v_t$ with $D$ (**a, b**) and $Z/E$ with $D$ (**c, d**) for the particle types plate-like (**a, c**) and side planes (**b, d**). $v_t$ graphs are all plotted for $p = 650\,\text{hPa}$ and $T = 255\,\text{K}$. See Tables A2 and A1 for the particle properties.

**Table 2.** Ratios between the results of the original and the disturbed retrieval. The row labeled mean gives the original input parameters, and the row labeled error is the range that has been used for variation.

| | Mean | Error | $D_m$ | $\mu$ | $E_1$ | $Z_1$ | $F_1$ | N and F<br>Scaled with E | N and F<br>Scaled with Z |
|---|---|---|---|---|---|---|---|---|---|
| p [pa] | 5.8E+4 | 5.0E+3 | 0.00 | 0.00 | 0.00 | 0.00 | -0.02 | 0.00 | 0.00 |
| T [K] | 248.15 | 10.00 | 0.00 | 0.00 | -0.03 | -0.01 | -0.04 | -0.04 | -0.02 |
| $\sigma_{total}$ [m s$^{-1}$] | 0.05 | 0.20 | 0.00 | 0.97 | 0.04 | 0.12 | 0.07 | 0.06 | 0.19 |
| $v_t$ [m s$^{-1}$] | 0.30 | 0.10 | 0.35 | 0.61 | 0.50 | 3.08 | 0.40 | 0.70 | 5.02 |
| | 0.90 | 0.10 | 0.10 | 0.20 | 0.20 | 0.50 | 0.10 | 0.28 | 0.82 |
| $Z/E$ [m mm$^{-6}$] | 7.5 | 5 | -0.11 | -0.25 | -0.18 | -0.42 | -0.20 | -0.26 | -0.68 |
| | 75 | 50 | -0.16 | -0.30 | -0.18 | -0.38 | -0.17 | -0.26 | -0.62 |
| Width [m s$^{-1}$] | 0.15 | 0.05 | 0.00 | -0.46 | -0.12 | -0.17 | -0.13 | -0.16 | -0.28 |

| Uncertainty factor |
|---|
| 0.00 … 0.10 |
| 0.10 … 0.30 |
| 0.30 … 1.00 |
| 1.00 … |

a factor of 3.0 uncertainty for $Z_1$ and 0.5 for $E_1$. These results are significantly different for a larger $v_t$ of $0.9\,\text{m s}^{-1}$ which produces uncertainties of 0.5 and 0.2 for $Z_1$ and $E_1$, respectively. The relatively large uncertainty of 70 % in $Z/E$ yields comparable lower uncertainty factors (0.4 for $Z_1$ and 0.2 for $E_1$). The relative errors derived for $N$ and $F$ are nearly identical because they are both derived from the same retrieved PSD. The question of whether the PSD is scaled via $E$ (error of $\pm 40\,\%$) or $Z$ (error of $\pm 2\,\text{dB}$) makes a large difference for the retrieval uncertainties $N$ and $F$. However, one has to keep in mind that the actual cause for the uncertainties $N$ and $F$ are the uncertainties in the underlying PSD.

In this analysis, only methodological errors and random measurement errors have been assumed. Also the influence of uncertainties in the calculation of fall speeds and choice of particles are left out of the estimation of uncertainties here. The assessment of their influences is actually not straightforward. Heymsfield and Westbrook (2010) found out that the

method of Mitchell (1996) performs especially well for pristine ice crystals with observed differences in a terminal fall velocity of 20 % at its maximum. For rimed ice crystals the error tends to increase significantly. Since conditions under which rimed particles occur are left out of this study, this can be ruled out as a major source of error. Also, the introduction of an additional exponent in the calculation of the Best number (as proposed by Heymsfield and Westbrook, 2010) does not change the results significantly. Uncertainties from the choice of particle type are also difficult to quantify. As long as the particles with a similar aspect ratio are chosen, the differences in the retrieved $N$ are around 50 %. As soon as a completely wrong type is chosen, e.g., columns instead of dendrites, the difference can be arbitrary because often only a very exotic solution or just no solution is found.

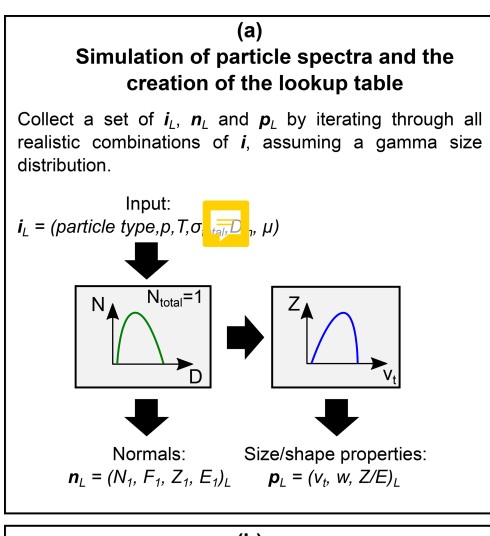

**(a)**
**Simulation of particle spectra and the creation of the lookup table**

Collect a set of $i_L$, $n_L$ and $p_L$ by iterating through all realistic combinations of $i$, assuming a gamma size distribution.

Input:
$i_L = (particle\ type, p, T, \sigma_{total}, D_m, \mu)$

N $N_{total}=1$ Z

D $v_t$

Normals:     Size/shape properties:
$n_L = (N_1, F_1, Z_1, E_1)_L$    $p_L = (v_t, w, Z/E)_L$

**(b)**
**Look up a result**

Create space with a combination of the coordinates (here: $v_t$, and $w$) and fill with the corresponding vectors $n_L$ and $i_L$.
Calculate the distribution of the matching probability in ($v_t$, $w$) space against vector $m=(v_{t,M}, w_M)$ measured with errors.

w

$e_2$
$w_M$

Coordinate of most-probable match

$e_1$   $v_t$
$v_{t,M}$

**(c)**
**Scale normal vectors and combine with P**

Retrieve vectors $r_L$ of extensive properties by scaling each normal vector of the lookup table with measured $Z_M$ and the simulated $Z_1$ so that $r_L = n_L (Z_M / Z_1)$.

Plot an element of all vectors $r_L$ vs matching probability $P$ (example for number concentration).

P

Uncertainty

N

Most-probable result

**Figure 6.** Step-by-step description of the retrieval principle. **(a)** A lookup table with parameters observable with lidar and radar is created with a set of combinations of particle shape, $p$, $T$, $\sigma_{total}$, $D_m$ and $\mu$. **(b)** The matching probability between the measured parameters and the simulated parameters is searched in the lookup table, resulting in estimations of $D_m$ and $\mu$. **(c)** The PSD created with the retrieved estimations of $D_m$ and $\mu$ is scaled with extensive parameters measured with lidar or CR ($E$ or $Z$, respectively). The full width half maximum of the distribution of matching probabilities is considered the uncertainty of the retrieved results. More explanation is given in the text.

## 4 Results – case study

Figure 2 shows an altocumulus cloud observed during the COLRAWI-2 campaign at Lindenberg on 11 June 2015. The observed cloud top temperature (CTT) is $-10\,°C$. The measured CR LDR (linear depolarization ratio) let us conclude that the ice particles present are most probably isometric ice crystals (Myagkov et al., 2016). Accordingly, the particle type "plate-like" is chosen for retrieving the microphysical parameters. Technically it would be possible to set particle type also as a retrieved parameter. However, the only measured parameter directly sensitive to particle shape would be cloud radar LDR, which is difficult to forward model and only rarely measured throughout the cloud case due to limitations in instrumental sensitivity. To avoid ambiguous results, we analyze the particle type manually and set it fixed for the whole cloud case. Otherwise the retrieved results would alternate between different particle types, introducing additional complications into the analysis.

The retrieval is done with three forms of the measurement vector which is used for the retrieval.

In the *first mode*, Z is the scaling variable and $m = (v_t, w)$ with a fixed error vector $e = (0.15\,m\,s^{-1}, 0.1\,m\,s^{-1})$. In this mode, only RWP and CR are used. $v_t$ is measured with the method of Radenz et al. (2018), and attenuation-corrected CR measurements of $Z$ and $w$ are also used for the retrieval. $p$ and $T$ are acquired from the European Center for Medium-Range Weather Forecast Integrated Forecast (ECMWF IFS) dataset. Figure 8a shows the result of the lookup-table-based retrieval with the measurement vector $m = (v_t, w)$ and Z as the scaling variable. The retrieval shows plausible results for $D$, $\mu$, $N$ and $F$ with about half an order of magnitude uncertainty, which is relatively constant all over the case. The large uncertainty results from a very broad probability distribution due to a stronger change at low values of $v_t(D_m)$ for this particle type (see Fig. 5) at low fall velocities.

In the *second mode*, all parameters are the same, except that $m = (Z/E, w)$ and $e = ((\Delta Z + \Delta E) \cdot Z/E, 0.1\,m\,s^{-1})$ with relative errors $\Delta Z = 0.2$ and $\Delta E = 0.1$. Extinction is derived by multiplication of the lidar backscatter $\beta$ with a lidar ratio of $32\,sr^{-1}$ which is typical for ice crystals (Haarig et al., 2016), using the Klett–Fernald approach (Seifert et al., 2007). In the latter case, only lidar and cloud radar are used for retrieval. Figure 8b shows the results of this run.

A *third mode* with $m = (Z/E, v_t, w)$ is presented in Fig. 9. Again, all other setting and parameters are the same. In this mode, a lot of pixels do not show a solution. Yet, for those where a solution is found, the magnitude of the results derived with this combined measurement vector are essentially the same as with the measurement vectors $(v_t, w)$ and $(Z/E, w)$. This is plausible because in the first two modes, noise from random variations in $Z$, $E$ and $v_t$ cannot be identified. The combined mode, taking into account all three measurement values, leads to a selection of only those results where the retrievals based on $Z/E$ and $v_t$ fit together.

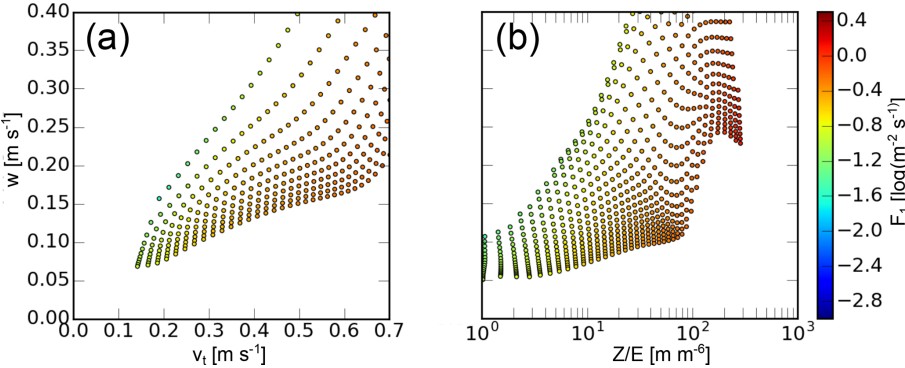

**Figure 7.** Example of a retrieval of $F_1$ based on $\boldsymbol{m} = (v_{\mathrm{t}}, w)$ **(a)** and $\boldsymbol{m} = (Z/E, w)$ **(b)** for the combined particle type plate-like.

**Figure 8.** Results of the retrieval based on $v_{\mathrm{t}}$ and $w$ (left column) and $Z/E$ and $w$ (right column) for the particle type plate-like. Combined CR/RWP measurements are only available in the second half of each hour. Different retrieved parameters are shown: **(a, h)** spectral width parameter $\mu$, **(b, i)** median diameter $D_{\mathrm{m}}$, **(c, j)** number concentration $N$ and **(d, k)** particle number flux $F$. Maximum probability of $P$ for each retrieval **(e, l)** is shown together with the upper **(f, m)** and lower **(g, n)** uncertainty factors for the retrieved parameters.

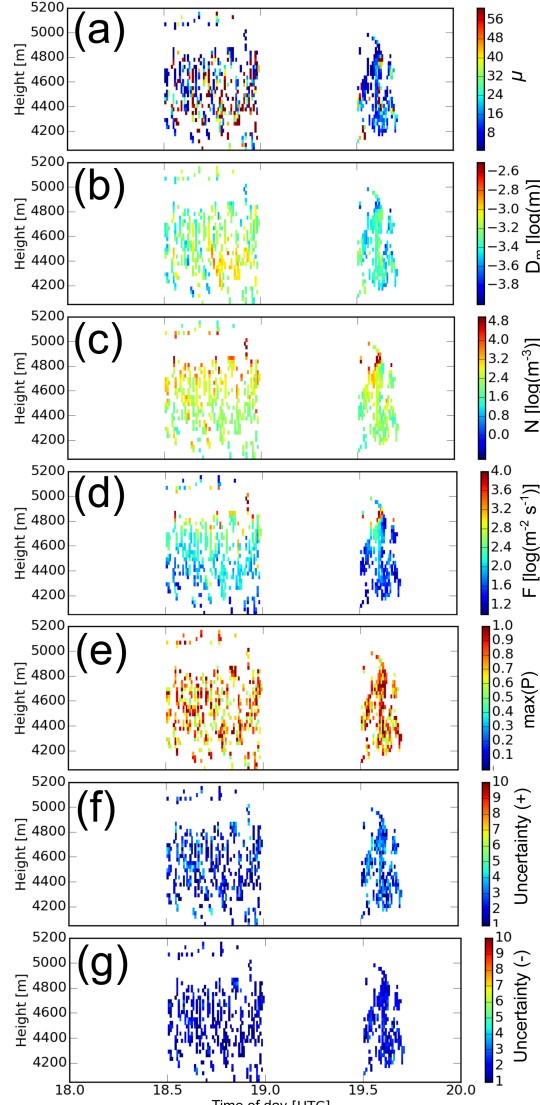

**Figure 9.** The same as Fig. 8 but with the combined approach taking into account $Z/E$, $v_t$ and $w$.

In all three modes, uncertainties are derived from the distribution of $P$ just as indicated in Fig. 6. As a means of quality control, values are only taken into account if $P > 0.9$ meaning that the retrieval is within the limits and a realis-
5 tic solution is found. Representative uncertainties of $N$ and $F$ are listed in Table 3 and a factor of 4 is used for the first mode; about a factor of 2 is used for the second; and below a factor of 2 is used for the third mode. Histograms of the retrieved $N$ and $F$ values for all modes are given in Fig. 10.
10 Averaged and median results for $N$ and $F$ for all four combinations of particle types and retrieval methods are also given in Table 3 for the height of 4500 m. At this single height, particle properties can be assumed constant in order to avoid additional stray of the retrieval caused by natural
15 variations. It is obvious that a low number of large outliers

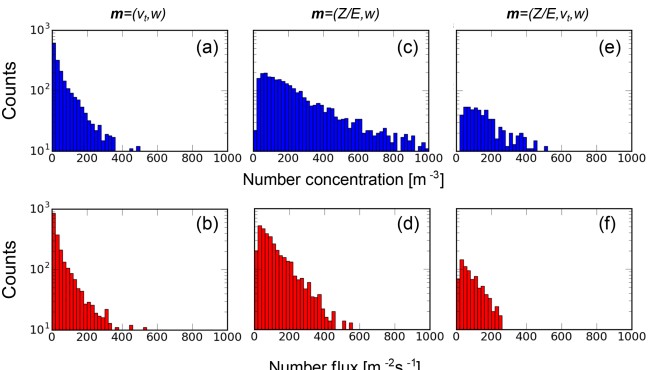

**Figure 10.** The distribution of retrieved results for $N$ and $F$ with the first (**a, b**), second (**c, d**) and third retrieval modes (**e, f**). The corresponding measurement vector **m** for each retrieval mode is indicated in the top row.

influence the average, so the median is better suited for a comparison of the results in this context showing mostly the intrinsic errors of the method. Based on the uncertainties of the individual retrieval results shown in Table 3, the results agree within 1 standard deviation of uncertainty. The his- 20 tograms of $N$ and $F$ show the best agreement between the second (CR/lidar) and third run mode (CR/lidar/RWP). The first mode (CR/RWP) seems to have a cutoff at $N = 400\,\mathrm{m}^{-3}$ and seems to tend more toward lower number concentrations. A small bias ($\delta v_t \approx 0.05\,\mathrm{m\,s}^{-1}$) could lead to such a tendency 25 enforcing larger particles and, hence, smaller number concentrations. A positive bias in cloud radar, in turn, would result in values that are too large in the second run mode.

## 5 Conclusions and discussion

A method has been demonstrated to retrieve the size and 30 shape parameters of PSDs from a combination of $E$, $Z$, $w$ and/or $v_t$. All combinations of the measured parameters have been tested and are found to produce results within the same order of magnitude. Uncertainties of the retrieval based on $v_t$ are found to be larger than for the estimation based on $Z/E$. 35 The smallest uncertainties are found if all measurement values ($E$, $Z$, $w$ and $v_t$) are taken into account; however, this is at the cost of a lower number of retrieved results. The largest retrieval uncertainties occur at smaller $v_t$ values for which the change of $D$ with $v_t$ is especially strong. 40

The method has its limitations only in the signal thresholds of the instruments (CR, lidar and/or RWP) and the a priori knowledge about the shape of the observed particles. In the current study, the CR has a signal threshold of about $-55\,\mathrm{dBZ}$ at a height of 5 km (Görsdorf et al., 2015); the 45 Polly$^{\mathrm{XT}}$ lidar can make useful measurements of cloud particle extinction down to about $50\,\mathrm{Mm}^{-1}$ (Bühl et al., 2013). The RWP is able to derive direct measurements of vertical velocity up to 6 km (Radenz et al., 2018). Since the latter

**Table 3.** Means, median and errors for the retrieval of $N$ and $F$ for the case study for different versions of $\boldsymbol{m}$ at 4500 m height.

| Measurement vector $\boldsymbol{m}$ | Mean $N$ | Median $N$ | Mean $F$ | Median $F$ | Averaged upper/lower uncertainty factors |
|---|---|---|---|---|---|
| $(v_t, w)$ | 2700 | 80 | 300 | 40 | $+4/-4$ |
| $(Z/E, w)$ | 2500 | 360 | 300 | 130 | $+2/-1.5$ |
| $(Z/E, v_t, w)$ | 700 | 382 | 190 | 150 | $+1.3/-1.2$ |

limitation is due to Bragg scattering it is essentially independent of the other instruments' signal thresholds. Hence, given a detection of vertical velocity by the RWP, the method presented here can be used to measure $N$ down to about 10–100 particle m$^{-3}$ (assuming a median diameter of about 800 μm) at a range of 5 km based on the sensitivity of the cloud radar. Such particle concentrations are usually too small to be detected by lidar; hence, the method presented here has an advantage over existing methods combining only lidar and radar.

The paper presents a first step towards the usage of the unique direct measurements of $v_t$ of pristine ice particles for the retrieval of their size, shape and number concentration. The forward modeling method used here for derivation of the observable measurement parameters is simple, but any other methods for CR forward modeling (Kollias et al., 2011) may be applicable for the computation of the lookup table. The present work also shows that it might be beneficial to use $v_t$ derived from combined CR/RWP measurements in other forward-iterating models like those of Ceccaldi et al. (2013) and Cazenave et al. (2019).

The presented method is essentially applicable to all remote-sensing facilities that provide a lidar particle extinction coefficient and CR (e.g., from the European Aerosols, Clouds and Trace Gases Infrastructure or the US program for Atmospheric Radiation Measurement). Combinations of lidar and radar prove most robust in terms of retrieval uncertainties. However, an error of less than 0.1 m s$^{-1}$ also allows for keeping methodological uncertainties for a retrieval of $N$ and $F$ in the range between a factor of 0.5 and 3, which brings the method to the edge of applicability. The method is crucial for the investigation of nucleation and the growth of ice particles in optically thick clouds for which no other method can provide accurate estimations of $N$ and $F$.

The forward model used here is transparent and instructive, but other forward-iteration methods might be used in the future as well. In the context of this work, the lookup table approach is used primarily for an analysis of retrieval uncertainties due to input measurement errors. Typing of the pristine particles on the basis of radar depolarization measurements is crucial for the methodology presented here. It is currently a field of intensive research (Bühl et al., 2016; Myagkov et al., 2016) and is, hence, left out of this work. The same applies to the accurate calibration of the CR which is a necessary prerequisite for the technique presented here

(Ewald et al., 2017), but it is also not covered in the present work.

Several issues need a solution for successful application of the method in future.

- Automatic particle typing must be improved. Recently developed methods employing scanning techniques (Myagkov et al., 2016) are more sophisticated and especially show better performance under low-signal conditions. Those techniques were not yet available during the COLRAWI-2 campaign, but they pose great opportunities for future application in the context of the present work.

- Uncertainties in CR calibration have not been taken into account here because those errors are essentially unknown. However, great effort is being made to come up with a solution for this problem.

- Matching between the CR and lidar beam has to be improved in order to avoid artifacts under complex situations.

- Direct information about local turbulence has to be taken into account to avoid errors in the estimation of the shape parameter $\mu$.

Given the downside of less flexibility, there are distinct advantages to the lookup table approach over classic forward-iteration methods; e.g., all possible results within the uncertainty range of the input variables are found at once. There is no risk that the method gets stuck in a local minimum. A lookup table approach also has the distinct advantage that numerical forward modeling and the actual retrieval are fully separable. Challenges to the approach are the extensive memory needs and the need for more effort in the evaluation of the results. The method is transparent, and it can be implemented easily in a numerically very efficient way. The computation of a case study as shown, e.g., in Fig. 8, takes less than a second on a state-of-the-art desktop computer including the estimation of retrieval uncertainties.

*Data availability.* The cloud radar data used in this work are available via the database of the Aerosol, Clouds and Trace Gases Research Infrastructure (ACTRIS) at https://actris.nilu.no (last access: 23 November 2019). Data from the Polly$^{XT}$ lidar and radar wind profiler are available from the authors.

## Appendix A: Mass and area power law relationships

Table A1 gives the parameterizations of $m$ and $A$ for different particle types according to Mitchell (1996). All values are given in the CGS (centimeter–gram–second) notation
5 system. Conversion to the SI system is done with the formula

$$\alpha_{SI} = \alpha_{CGS} \cdot 100^{\beta_{CGS}}/1000, \tag{A1}$$
$$\gamma_{SI} = \alpha_{CGS} \cdot 100^{\sigma_{CGS}}/10\,000. \tag{A2}$$

The lookup table used in this paper is computed of all particle types with a single parameterization from Table A1. Additional particle shapes are defined in Table A2 that include a 10 transition to aggregate particle types at the upper limit of $D$.

**Table A1.** Values for $\alpha$, $\beta$, $\gamma$ and $\sigma$ as well as their valid ranges ($D_{min} \ldots D_{max}$) used in the context of this work.

| Particle type | $D_{min}$ | $D_{max}$ | $\alpha_{CGS}$ | $\beta_{CGS}$ | $\gamma_{CGS}$ | $\sigma_{CGS}$ |
|---|---|---|---|---|---|---|
| Hexagonal plates | 0 | 99 | 0.0065 | 2.45 | 0.24 | 1.85 |
| | 99 | 400 | 0.00739 | 2.45 | 0.65 | 2 |
| Hexagonal columns | 0 | 99 | 0.1677 | 2.91 | 0.684 | 2 |
| | 99 | 300 | 0.00166 | 1.91 | 0.0696 | 1.5 |
| | 300 | 600 | 0.000907 | 1.74 | 0.0512 | 1.414 |
| Rimed long columns | 600 | 2000 | 0.00145 | 1.8 | 0.0512 | 1.414 |
| Sector branched crystal | 0 | 40 | 0.00614 | 2.42 | 0.24 | 1.85 |
| | 40 | 8000 | 0.00142 | 2.02 | 0.55 | 1.97 |
| Broad branched crystal | 0 | 100 | 0.00583 | 2.42 | 0.24 | 1.85 |
| | 100 | 1000 | 0.000516 | 1.8 | 0.21 | 1.76 |
| Stellar crystal broad arms | 0 | 90 | 0.00583 | 2.42 | 0.24 | 1.85 |
| | 0 | 1500 | 0.00027 | 1.67 | 0.11 | 1.63 |
| Densely rimed dendrite | 0 | 4000 | 0.015 | 2.3 | 0.21 | 1.76 |
| Side planes | 0 | 2500 | 0.00419 | 2.3 | 0.2285 | 1.88 |
| Bullet rosettes | 0 | 1000 | 0.00308 | 2.26 | 0.0869 | 1.57 |
| Aggregates side planes | 0 | ~~1 000 000~~ | 0.0033 | 2.2 | 0.02285 | 1.88 |
| ~~Aggregates thin planes~~ | 600 | 1 000 000 | ~~0.00145~~ | ~~1.8~~ | ~~0.2285~~ | ~~1.88~~ |
| Aggregates mixture | 0 | 8000 | 0.0028 | 2.1 | 0.2285 | 1.88 |
| Assemblage planar polycrystals | 0 | ~~1 000 000~~ | 0.00739 | 2.45 | 0.2285 | 1.88 |
| Lump graupel | 500 | 3000 | 0.049 | 2.8 | 0.5 | 2 |
| Hail | 5000 | 25 000 | 0.466 | 3 | 0.625 | 2 |

**Table A2.** Combined particles types used in the context of this work.

| Combined type no. | Combined type name | Parameterizations | Ranges |
|---|---|---|---|
| 1 | Plate-like | Hexagonal plates | 15–600 μm |
| | | Aggregates mixture | 600–3000 μm |
| 2 | Column-like | Hexagonal columns | 30–400 μm |
| | | Rimed long columns | 600–2000 μm |

## Appendix B: Calculation of terminal fall velocity

This appendix gives a detailed description for the actual calculation of $v_t$ of particles with known parameterizations of particle mass $m$ and area $A$. It is mentioned here for the sake of completeness. We follow here the method described in Khvorostyanov and Curry (2014).

$$m(D) = \alpha D^\beta \tag{B1}$$
$$A(D) = \gamma D^\sigma \tag{B2}$$

$$g = 9.81 \, \mathrm{m \, s}^2$$

The density of the air,

$$\rho_{\mathrm{air}} = p/(R_{\mathrm{air}} T), \tag{B3}$$

with $R_{\mathrm{air}} = 287.058 \, \mathrm{J \, kg}^{-1} \, \mathrm{K}^{-1}$ and kinematic viscosity,

$$\nu = \eta_{\mathrm{air}}/\rho_{\mathrm{air}}, \tag{B4}$$

with $\eta_{\mathrm{air}} = 1.59e-5 + (1.725e-5 - 1.59e-5) \cdot (T - 250.0)/25.0 \, \mathrm{kg \, m}^{-1} \, \mathrm{s}^{-1}$ are used to derive the Best number,

$$X(D) = 2m_{\mathrm{D}} \left(1 - \rho_{\mathrm{air}}/\rho_p\right) g D^2 / \left(A_{\mathrm{D}} \rho_p \nu^2\right). \tag{B5}$$

The two constants

$$b_{\mathrm{Re}}$$
$$= C_1 X^{0.5} / \left(2\left(\left(1 + C_1 X^{0.5}\right)^{0.5} - 1\right)\left(1 + C_1 X^{0.5}\right)^{0.5}\right), \tag{B6}$$

$$a_{\mathrm{Re}} = \delta_0^2/4.0 \left(\left(1 + C_1 X^{0.5}\right)^{0.5} - 1\right)^2 \Big/ X^{b_{\mathrm{Re}}} \tag{B7}$$

with $C_1 = 4/(\delta_0^2 C_0^{0.5})$ are defined. The different constants that apply for different particle types are defined in Table B1. The Reynolds number,

$$Re(D) = a_{Re}(D) X(D)^{b_{Re}(D)}, \tag{B8}$$

is derived and two additional functions are defined,

$$A_v = a_{Re} \nu^{1-2b_{Re}} (2\alpha g/(\rho_{\mathrm{air}}\gamma))^{b_{Re}}, \tag{B9}$$
$$B_v = b_{Re} \cdot (\beta - \sigma + 2) - 1. \tag{B10}$$

With these results, $v_t$ can be expressed as a closed function of $D$.

$$v(D) = A_v(D) \times D^{B_v(D)} \tag{B11}$$

**Table B1.** Parameters for velocity calculations (Böhm, 1989).

| Particle type | $C_0$ | $\delta_0$ | $\rho_p$ [kg m$^{-3}$] |
|---|---|---|---|
| Ice crystals | 0.6 | 5.83 | 934.0 |
| Hail/graupel | 0.292 | 9.06 | 934.0 |
| Rain/drizzle droplets | 0.292 | 9.06 | 1000.0 |

*Author contributions.* JB initiated the COLRAWI measurements, conceived the retrieval method and wrote the paper. PS and MR contributed methods for data evaluation and interpretation. HB evaluated the Raman lidar data. AA supervised the work and supported the preparation of the paper.

*Competing interests.* The authors declare that they have no conflict of interest.

*Special issue statement.* This article is part of the special issue "BACCHUS – Impact of Biogenic versus Anthropogenic emissions on Clouds and Climate: towards a Holistic UnderStanding (ACP/AMT/GMD inter-journal SI)". It is not associated with a conference.

*Acknowledgements.* We thank Volker Lehmann, Ulrich Görsdorf and Ronny Leinweber of MOL/RAO Lindenberg for their cooperation and for performing the measurements with RWP, CR and DL. The Polly$^{XT}$ used in this study is under the supervision of Ina Mattis (DWD).

*Financial support.* This research has been supported by the Deutsche Forschungsgemeinschaft (grant no. 398285025), the European Union's Framework Programme for Research and Innovation, Horizon 2020 (ACTRIS-2 (grant no. 654109)), the former European Commission Seventh Framework Programme FP7/2007–2013 (ACTRIS (grant no. 262254); BACCHUS (grant no. 603445)) and the European Union (Cloudnet (project no. EVK2-2000-00611)). CE1

The publication of this article was funded by the Open Access Fund of the Leibniz Association.

*Review statement.* This paper was edited by Johannes Schneider and reviewed by two anonymous referees.

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

**Remarks from the language copy-editor**

CE1    Please review this section carefully as there has been extensive rewording.

**Remarks from the typesetter**

TS1    Please check again the second reference (i.e. Böhm, 1992) you sent me. They seem to be identical.