# Peer review of "Ice crystal number concentration from measurements of lidar, cloud radar and radar wind profiler"

_Atmospheric Measurement Techniques, 2019_

## Referee Comment (RC1) · Anonymous Referee #1 · 30 Jun 2019

The derivation of ice crystal number concentration is an important task to better understand the effect on and off ice clouds. This paper presents a method to derive the ice crystal number concentration from a set of active remote sensing instruments and is thus highly suited for presentation in AMT. I suggest only a few minor points that could be addressed before publication of this article:

In you introduction you describe shortly current existing methods to derive ice crystal number concentration, i.e. using lidar and radar or multi-wavelength radar methods. You mention that each of this method has its limitations. In your method you use a combination of either CR+RWP or CR+lidar. What is the detection limit of the different instruments to detect clouds? Can you give an assumption of the benefit of the proposed method in comparison to the others mentioned here (number)? What about regions with very low ice optical thickness / very small ice crystals where only lidar measurements can detect the cloud? Isn't there still a limitation with this method to investigate ice crystal number concentration for these thin clouds. This should be addressed in the manuscript.

You do your study using side planes and column-like particles. How different would it look like, if other shapes are taken into account. Are these particle shapes enough to address all regions / stages of the cloud? What about small pristine ice particles?

The description of the results is very short.

The results for using CR+RWP and CR+lidar look quite different (Figure 8). Can you give a better comparison of the both methods. How do you deal with that in your studies. Do you use the different methods for different parts of the clouds? If yes, how is that done? If no, what about clouds in which none of the methods can be applied for the whole cloud?

Can you give a more quantitative comparison of the results using the different assumed particle shapes?

Figure 5: Please use the same range of D for both methods.

---

## Author Comment (AC1) · 20 Sep 2019

**Revision of "Ice crystal number concentration from measurements of lidar, cloud radar and radar wind profiler",**

*Johannes Bühl, Patric Seifert, Martin Radenz, Holger Baars, and Albert Ansmann*

*(Leibniz Institute for Tropospheric Research, Permoserstr. 15, 04318 Leipzig, Germany, buehl@tropos.de)*

We thank both reviewers for their thorough analysis and comments to our paper. Below we list our answers to these comments together with a description of the applied changes.

**Referee #1:**

The derivation of ice crystal number concentration is an important task to better understand the effect on and off ice clouds. This paper presents a method to derive the ice crystal number concentration from a set of active remote sensing instruments and is thus highly suited for presentation in AMT. I suggest only a few minor points that could be addressed before publication of this article:

In you introduction you describe shortly current existing methods to derive ice crystal number concentration, i.e. using lidar and radar or multi-wavelength radar methods. You mention that each of this method has its limitations. In your method you use a combination of either CR+RWP or CR+lidar. What is the detection limit of the different instruments to detect clouds? Can you give an assumption of the benefit of the proposed method in comparison to the others mentioned here (number)? What about regions with very low ice optical thickness / very small ice crystals where only lidar measurements can detect the cloud? Isn't there still a limitation with this method to investigate ice crystal number concentration for these thin clouds. This should be addressed in the manuscript.

The method presented here has its limitations in the detection limits of the single instruments. Detection limits of the cloud radar (-55dBZ at 5km) and that of the lidar (about 1e-6 Mm$^{-1}$) are now mentioned in the discussion section (p. 19 l. 1).

For the combination of lidar/cloud radar methods exist and are well studied. In principle, any of the recent algorithms (e.g. DARDAR) could be run also with ground-based instrumentation. A specific advantage – and the actually new development presented in this paper – results from the combination of radar wind profiler and cloud radar. The method is limited by the capability of the cloud radar to detect particles and by the detection limits of the RWP concerning the velocity of clear air. Both detection limits are essentially independent. The RWP has a detection limit of about -30dBZ at 5km and about 90% coverage for wind measurements at 6km height. The detection limits of the cloud radar of -55dBZ at 5km height translates into a possible detection of about 10…100 columnar-shaped ice particles per cubic meter with 600…1000µm median Diameter. Those small particle concentrations would under most circumstances not be detected by a lidar, hence, it brings an advantage that the only extensive measurement parameter used by the method is Z from the cloud radar (given an additional detection of air velocity of the RWP).

We added a paragraph about this topic to the discussion (p. 16 l 25).

You do your study using side planes and column-like particles. How different would it look like, if other shapes are taken into account. Are these particle shapes enough to address all regions / stages of the cloud? What about small pristine ice particles?

Usage of a wrong ice particle shape usually ends up with no or a very exotic solution. E.g. if columnar ice particles are present with a relatively large terminal fall velocity of 1m/s, the retrieval with side planes would simply fail because these particles would never reach this terminal fall velocity in a pristine state.

Small pristine particles are best covered by the method, because the parameterizations (especially columns and hexagonal plates) work well down to very small sizes of about 10µm).

The description of the results is very short.

We added a new Figure showing the histograms of the retrieved distributions of $N$ and $F$ throughout the whole cloud case. These plots show the natural stray of the results within the cloud case and are mostly caused by changes in particle properties. The uncertainties of the individual measurement points are now shown in Figs. 8 and 9. Table 3 now gives a comprehensive overview about the resulting $N$ and $F$ and the corresponding errors for the height of 4500m.

We extended the results section to be more descriptive with a depiction of the retrieval results of $N$ and $F$ as a collection of histograms. Additionally, a new case study with the combined measurements of Lidar, CR and RWP was added. (See comment 3 of Reviewer #2).

The results for using CR+RWP and CR+lidar look quite different (Figure 8). Can you give a better comparison of the both methods. How do you deal with that in your studies. Do you use the different methods for different parts of the clouds? If yes, how is that done? If no, what about clouds in which none of the methods can be applied for the whole cloud?

We have intentionally selected a case study in which both methods are applicable. Later, we also want to apply the CR/RWP method to clouds in which Lidar is attenuated, but first we want to research the capabilities of both methods together. The most simple solution for the application in strongly changing conditions would be to have a sophisticated error analysis at all heights. Larger errors would then limit the influence of the corresponding measurement value to the retrieval and increase the overall errors of the retrieval. Currently, we regret that our error estimation is not yet sophisticated enough to yield usable errors for all heights automatically, so we still have to estimate the errors for the input variables.

Can you give a more quantitative comparison of the results using the different assumed particle shapes?

A comment on the errors resulting from different particle shapes was added to p.14 l. 7.

Figure 5: Please use the same range of D for both methods.

Thank you for this advice, we fixed the scale accordingly.

The manuscript describes a methodology that aims at estimating the ice crystal number concentration ($N_i$) and ice crystal number flux $F_i$ ($N_i$ multipled by the terminal fall velocity $v_t$) based on combined measurements from a ground-based cloud radar (CR) and a radar wind profiler (RWP) or lidar. More specifically, the authors present here two retrieval methods that can independently be used depending on instrumental availability. A first method uses the CR reflectivity ($Z$) and spectral width ($w$) and the RWP $v_t$ to constrain parameters of a particle size distribution (PSD) and infer $N_i$ and $F_i$. Alternatively, in the absence of RWP measurements, the lidar extinction ($E$) is instead used in combination with $Z$. The authors describe the theoretical basis for these methods, which are based on rather sophisticated LUT approach, and propose a brief uncertainty analyses. Finally, a case study representative of a mixed-phase cloud is discussed.

There is high scientific interest for this work, as only few methods are today dedicated to remote-sensing retrievals of $N_i$, despite the importance of this parameter to better understand ice clouds and represent them in models. A particular novelty of the approach proposed in this manuscript is its strong focus on radar measurements and the use of a RWP. Indeed, other existing methods for $N_i$ retrievals strongly depend on lidar and/or thermal infrared measurements and therefore perform poorly in optically thick ice clouds. The idea presented here is thus very interesting, but I still have a some major concerns regarding some aspects of the retrieval method, detailed in the comments below. Also, I would have liked to see more analyses of the retrievals, especially since the authors indicate in Sec.2 that measurements are available during a 4-month period and later in the conclusion that the retrieval method is very fast. But this paper already constitutes a first step and the manuscript is well within the scope of AMT. The manuscript is well written, although can be a bit difficult to follow in its most technical sections. Overall, I recommend publication of the manuscript after major revision, provided appropriate response to the comments below.

General comments:

1. My first general comment concerns the use of a LUT approach, which appears as a strong limitation here considering the great instrumental synergy that could be obtained here. I was wondering why this choice, until the very last paragraph of the conclusion where it is finally justified. There are clearly advantages of going for a LUT approach (keeping a clearer view of the physical aspects, retrieval speed), which are mentioned by the authors, but I can't help but think so much more could be done with a variational method: all measurements could be used

simultaneously, and a proper sensitivity / error estimates study could be performed (see my following two comments). Obviously, the authors shouldn't switch to a variational method for this study, but such discussion must be addressed earlier in the manuscript, and I strongly encourage the authors to migrate to a more flexible method in the future.

A sophisticated forward iteration approach would be an appropriate solution for the problem. We are actually working on that but so far we have to admit that we have no such approach ready. However, we see specific advantages of the LUT that we want to push forward and explore.

- We see strong advantages in the calculation speed and in simplicity in application and portability.

- The original idea behind the LUT approach was its ability to show all results at once. If the particle species are involved is a free parameter, there is a certain danger that a forward iteration gets stuck in a local minimum, neglecting possible other solutions nearby. Within one particle type that turned out not to be a problem and since we are currently selecting particle species manually this is no big issue in the current state of our work. However, if in future a forward iteration approach is used, the LUT may be able to provide a single or many starting points for the forward iteration.

2. I have issues with the error analysis done in section 3.5 and summarized in table 2. Uncertainties on retrievals should have two origins: first, the propagation of errors due to non-retrieved parameters (here p, T, s, particle shape, parameters of M(D) and A(D) relations, ...) and instrumental accuracy (here $v_t$, $Z/E$, $w$), and second the sensitivity of these measurements to the parameters to be retrieved. It seems that only the first aspect is addressed here? And, if so, why not include the errors due to the choice of a particle shape in Table2? Since this is also a non-retrieved parameter, it should appear there. It would also be good to attribute uncertainties on the assumed parameters of the mass-diameter and area-diameter relations. Finally, showing directly the relative uncertainties on $N_i$ and $F_i$ in table 2 would be clearer for the reader.

Technically it is possible to set particle shape as a retrieved variable. However, the only measured parameter directly sensitive to particle shape would be cloud radar linear depolarization ratio (LDR), which is difficult to forward model and only rarely measured throughout the cloud case due to limitations in instrumental sensitivity. To avoid ambiguous results, we analyze the particle type manually and set it fixed for the whole cloud case. Otherwise the retrieved results would alternate between different particle types, introducing additional complications into the analysis.

Errors originating from the mass and area parameterizations and the calculation of terminal fall velocities are difficult to asses. Heymsfield and Westbrook (2010) found out that the method of Mitchell (1996) performs especially well for pristine ice crystals with observed differences in terminal fall velocity below 20%. The distribution of these errors also seems to be randomly distributed. For rimed ice crystals the error tends to increase significantly. Since conditions under which rimed particles occur are left out of this study, this can be ruled out as a major source of error. Also introduction of an additional exponent into the calculation of Best numbers as proposed by Heymsfield and Wesbrook (2010) does not show any significant differences.

We added this sentence to the discussion of uncertainties (p. 14, l. 7 and to the result section (p. 16, l. 11ff). We also included the lidar and radar errors into the calculation of N and F now in table 2 (in the other estimations of uncertainties the measurement errors have already been included).

3. Why are the CR, RWP and lidar not all used together if they are available simultaneously.The authors seem to hint at a redundancy between the information in $v_t$ and $Z/E$, but the case study shows that should be different type of information. And it is also expected that the lidar ($E$) carries information on a different part of the PSD (small particles) compared to the CR and RWP. The authors indeed show in the case study that there is almost no difference between the mean $N_i$ and $F_i$ retrieved by both methods if the "side-planes" shape is used whereas very large differences between the two methods appear for "plate-like" particles. Doesn't this indicate that there is information on the particle shape that could be further constrained by simultaneously using all available measurements?

(See answer two question 2)

In any case, the discrepancies between the results from both methods should be further discussed.

It is actually possible to use the method in a combined way, using $\boldsymbol{m}=(Z/E, v_t, w)$ as an measurement vector. This kind of evaluation is now also shown and discussed in the results section. In this context we dropped the additional comparison of retrieval results between different particle types and concentrate the analysis on particle type "plate-like". This particle type is according to our knowledge the best choice for this case.

The most notable difference using $\boldsymbol{m}=(Z/E, v_t, w)$ is that the results stray less and the errors are smaller. Probably, in the Lidar/CR and CR/RWP approach random noise and biases in the measurements translate into a noisy but complete timeheight field of results with large errors. The combined approach will only select measurements in which *Z/E* and $v_t$ fit together.

We also discuss briefly the differences between the three retrieval modes in the results and discussion section.

4. Finally, it would be useful to explicit more clearly in the abstract or conclusion the conditions under which the retrieval methods can be applied. For instance, p8 l6: "only small pristine particles are considered in the context of this work" (and what does "small" mean?). Same p4 l17. Also, do these retrieval methods only apply to thick and relatively warm mixed-phase clouds as illustrated in the case study?

We removed "small" from the study and only mention "pristine" which better describes our priorities. (In principle, all particle with clearly defined shape for which an area and mass power law is known can be treated.)

This is now mentioned in the text.

Specific comments:

5. The study by Delano¨e et al. [2005] is cited as a reference to the shape of the PSD used in thisstudy (gamma-modified, Eq. 1), which is the most central element of the retrieval method. The authors also indicate, p3 l9-10, that "this kind of distribution was described by Delano¨e et al. [2005] as being universally applicable for ice crystal populations". First, a main conclusion by Delanoe et al. [2005] (or, more recently, Delano¨e et al. [2014]) is that, **when properly normalized**, PSDs tend to fold into a unique universal modified-gamma distribution shape. By normalized it is meant that the concentration and size axes of the PSD are normalized in order to remove dependency on parameters that are important to the shape (in the case of Delano¨e et al. [2005], it is IWC and $D_m$ that become constant after normalization). I do not see mention of this central aspect in the text. It seems from Eq. 1 that D is indeed normalized by Dm and that there is a normalization coefficient for the concentration (*C*), but the latter isn't discussed anymore. Please comment. Second, the PSD shape in Eq.1 does not correspond to a gamma-modified PSD, but it does correspond to a gamma-$\mu$ indeed mentioned in Delano¨e et al. [2005] but that was found to be a less accurate representation of in situ PSDs than the gamma-modified shape. Please clarify.

We use the gamma-µ distribution because it allows control about both the shape and the size variable with two free parameters. We refrained from using the gamma-modified distribution because there were three free parameters in it. But from the perspective of your comment we think, usage of a gamma-modified

distribution with some fixed parameters would also make sense. We integrated it as an option to the software which will be published with the manuscript. Meanwhile, we also see that the shape parameter of the distribution seems only to play a major role under extreme conditions. So, indeed, switching to a gamma-modified distribution could be a good option to make the algorithm more stable.

The introduction was modified accordingly (p. 3 l. 10), the gamma-μ distribution is now correctly named and referenced and the normalization parameter $C$ is defined properly (p. 6 l. 21).

6. p2 l11, and onward in the paragraph - there are many mentions of "crystal size", please define more clearly what you mean by that (maximum size, effective size?). Or do you mean the PSD? This confusion happens often in the text. For instance, p9 l19,"ice particle size D is the most crucial intensive parameter for the retrieval of N" - is it D or N(D) that is crucial to N (I'd agree on the latter)? Similarly, in p3 l8 there is mention of a median diameter $D_m$, could you explicit what you mean by this? As Delanöe et al. [2005] is mentioned, is it the mean volume-weighted diameter (ratio of 4th to 3rd moment of the PSD)?

D is always the maximum Diameter of a single particle and Dm the median Diameter of the particle size distribution.

We tried to make this clear throughout the text. $D$ is now described at p. 4 l. 19. Wherever applicable we replaced "particle size" by the corresponding symbol ($D$ or $D_m$). "particle size distribution" was replaced by "PSD" throughout the text.

7. p2 l13: Ceccaldi et al. [2013] describes a method to infer a cloud classification (phase) fromlidar-radar measurements, this is not an appropriate reference here (again p16 l3). If you refer to the DARDAR-CLOUD algorithm, then perhaps Cazenave et al. [2019] or Delanöe and Hogan [2010] would be appropriate, but none of these papers discuss of $N_i$ retrievals either. As far as I know, only Mitchell et al. [2018] and Sourdeval et al. [2018] describe satellite products of $N_i$. The former paper uses thermal infrared and lidar measurements, the latter uses lidarradar measurements. Additionally, Sourdeval et al. [2018] showed that $N_i$ retrievals using lidar measurements only are possible for cold ice clouds, so the statement p2 l14 is not absolutely correct. However, it indeed seem that their satellite retrievals are indeed poor when the lidar is missing.

Thank you for this insight. We tried to improve the introduction according to these comments with updated descriptions of the cited papers. Additionally, we cite Mitchell et al. (2018).

8.  Appendix B lacks references or details. Some are given p9 l21-22, and that seems sufficientto justify Eq.9, so perhaps Appendix B is not necessary?

We think Appendix B is important for the sake of completeness and for easy reproducibility of our method. It shows how to actually calculate the terminal fall velocities which are central to this work.

We now introduce the chapter with references to the relevant papers.

9.  Table A1 lists m(D) and A(D) coefficients for all sorts of shapes, many of which aren't used.Is it useful to indicate them all?

We mention all possible shapes for the sake of completeness, since there is no principle limit in application of the method as long the particle shape is known.

10. $n_L$ and $i_L$ are inconsistent between the equations p10 and Fig.6. Also, in Fig6b, what are the space coordinates $x_i$?

The coordinates were designated $x_i$ in order to have a general approach. We decided to drop this idea and make the coordinate system specific in order to be consistent between the figures and the text.

Technical comments:

11. p6 l2 - typo, "cloumn-like"

Done

12. p6 l4 - "particle shapes" rather than "particle species"?

Done

13. p9 l19 - type, "parameter"

Done

14. p9 l28 - wrong citation format

Exchanged (Fukuta and Takahashi, 1999) to Fukuta and Takahashi (1999)

15. p10 l19 - D or Dm?

   Exchanged D → D$_m$

16. p13 l2 - bold m (vector) or $m_i$ (element)? There are other similar inconsistencies in this section.

   Exchanged *m* to **m** and *e* to **e** at all occurrences of the measurement vector / error vector.

17. p13 l1 - can you define "precisely"? What are exactly the elements of *m*? Are p and T included?

   We reformulated the paragraph in order to explain that the vector can hold all variables, but later only v$_t$, w and Z/E are used.

18. p13 l15 - does "the desired vector r" corresponds to the retrieved properties? (understood from Fig. 6 but not detailed in the text)

   We defined vector $r$ in the text-

19. Fig. 5 - is it D or Dm on the y-axis?

   It is indeed D$_m$

**Additional changes due to internal review:**

- Table A2: aggregates thin plates → aggregate mixtures (This was a typo. In the analysis always the "aggregate mixtures" type has been used for parameterizing the upper end of the plate-like crystals because the fall velocity parameterization matches best with "hexagonal plates".)

- Plot color scales N1 → N, F1 → F

- A reference to Hogan et al. (2006) has been added after Eq. 1.

- Symbol $s$ has been exchanged to $\sigma_{total}$ in the whole paper.

[revised manuscript text omitted]